# Retained Singular Values in Probabilistic Image Segmentation with Optimal Transport

## Abstract

Latent probabilistic models are a popular choice for quantifying aleatoric uncertainty in image segmentation tasks. This uncertainty is typically modelled as latent axis-aligned Normal densities. However, we find that the singular values of the modeled densities can vanish and result in a poor, inexpressive latent space. Deterministic self-supervised models have achieved state-of-the-art results by optimizing embeddings on a projected space to successfully retain the latent singular values. In this work, we extend this approach to the probabilistic setting by introducing the Conditional Sinkhorn Auto-encoder (cSAE). It is shown that with Normalizing Flows and Optimal Transport theory, we can project the latent space and improve the learned embeddings of supervised conditional probabilistic segmentation models. We show that this is due to the singular values of the learned Normal densities being better retained, thereby improving the ability to accurately model and quantify the data uncertainty.

## 1 Introduction

A good understanding of the input data is important for developing successful deep learning models. Data is almost always obtained with inherent ambiguity, which is commonly referred to as the *aleatoric uncertainty*. Ignorance of this uncertainty can result in an incorrect interpretation of model predictions. In the context of vision, ambiguity in images can be due to occlusions, shadows, sensor noise, insufficient resolution or other equivocal information.

Probabilistic modeling can enable learning of the aleatoric uncertainty in the data. Kohl et al. (2018) introduced a VAE-based model, the Probabilistic U-Net (PU-Net), to learn latent embeddings of the segmentation ambiguity as amortized axis-aligned Normal densities. This model has been the most popular backbone for aleatoric uncertainty quantification in segmentation (Hu et al. (2019); Kohl et al. (2019); Valiuddin et al. (2021); Selvan et al. (2020); Baumgartner et al. (2019); Mehrtash et al. (2020). Nevertheless, we show that the current model is sub-optimal and provide evidence that suggests that the relative magnitude of its singular values play a role in the segmentation performance of the PU-Net. This work introduces the phenomena of vanishing singular values in the PU-Net and possibly other similar models. This problem occurs due to the learned latent prior density, which is by default fixed throughout training in the conventional VAE. This can be a significant bottleneck for probabilistic segmentation, where variance in the latent densities is essential to properly encapsulate the aleatoric uncertainty. We propose an improved model, the Conditional Sinkhorn Autoencoder (cSAE), that retains the singular values by using Normalizing Flows (NFs) and Optimal Transport (OT) in the latent space.

We highlight the resemblance between the PU-Net and joint embedding methods (JEMs) in self-supervised learning (SSL). Where in both models, two networks attempt to learn their semantically relevant mutual information subject to changes in the representation (e.g. data augmentation). In these works (Wu et al. (2018); Bachman et al. (2019); Chen et al. (2020)), so-called projection layers are utilized to avoid the vanishing of the singular values (Jing et al. (2021)). We argue that the low-variance singular values of the Normal densities limit probabilistic segmentation performance. Thus, we attempt to solve this problem in a similar fashion as to what is done with the deterministic projection layers in JEMs. Instead of using a

multi-layered perceptron, we utilize Normalizing Flows for probabilistic projection. Normalizing Flows have been used in previous works (Selvan et al. (2020); Valiuddin et al. (2021)) to improve the latent space of the PU-Net. However, in these works, only the posterior density is projected with an NF and little intuition is provided on why this improves performance. Our work aims to project both latent spaces and show it can prevent the singular values from vanishing.

In summary, we argue that the PU-Net and other similar models with learned priors do not fully utilize their latent space due to its inherent learning mechanism. Furthermore, we demonstrate that shifting from minimizing the Evidence Lower Bound to the perspective of minimizing the Optimal Transport plan results in better retained singular values of the Normal densities and as a result, a more expressive latent space. In Section 2, we discuss related work regarding probabilistic segmentation, projection layers in self-supervised learning and generative modeling with Optimal Transport. In Section 3, we present the theory on Normalizing Flows (Section 3.1), the Variational Autoencoder (Section 3.2) and Optimal Transport (Section 3.3). Consequently, we present our methods in Section 4, where we combine the aforementioned theoretical frameworks. Then, the results are discussed in Section 5. Here, it can be seen that our proposed approach performs significantly better under various test evaluations and produce more accurate segmentations. Finally, we conclude our work in Section 6.

## 2 Related work

### 2.1 Probabilistic segmentation with Normalizing Flows

Kohl et al. (2018) introduced the PU-Net for image segmentation by combining the conditional Variational Autoencoder (cVAE) of Sohn et al. (2015) and the U-Net of Ronneberger et al. (2015). Additionally, the authors allow the conditional prior density to be learned, instead of fixing it throughout training. This enables the model to provide multiple plausible segmentation hypotheses per image, and as a result, capture the ambiguity in the data. The PU-Net has received significant attention and various improvements due to increasing interest in methods for quantifying aleatoric uncertainty. Hu et al. (2019) use the inter-observer variability as a target in the training objective. Selvan et al. (2020) and Valiuddin et al. (2021) use NFs to augment the posterior network, originally introduced by Rezende & Mohamed (2015) in the non-conditional setting. Valiuddin et al. (2021) hypothesize that the improvement in performance is due to the non-Gaussian nature of the posterior density enabled by the NF augmentation. While this can certainly be true, we argue in this work that a major contribution to the increased performance can be accredited to the regularizing effect of the augmented projection space.

### 2.2 Dimensional collapse in self-supervised learning

Jing et al. (2021) have investigated the effect of the commonly used multi-layered perceptron head (also referred to as a *projector*) in JEMs (Chen et al. (2020); Bachman et al. (2019); Wu et al. (2018)). Empirically, it is found that a simple projection layer retains the singular values of the preceding layer and results in better embeddings. Furthermore, they argue that the projection layer reduces the degree of dimensional collapse caused by implicit regularization (Ji & Telgarsky (2018); Gunasekar et al. (2017); Arora et al. (2019)) due to over-parameterization (Saxe et al. (2019); Neyshabur et al. (2018); Soudry et al. (2018); Barrett & Dherin (2020)) and strong data augmentation. Chen et al. (2020) have motivated the importance of projection heads by pointing to the induced information loss caused by the training objective. We suggest that the posterior-augmented NF serve a similar purpose to that of the projection layers in JEMs and indeed retain the singular values of the prior latent densities. Both the PU-Net and JEMs attempt to learn the mutual information of the input images. The encoders of the PU-Net capture the mutual information between the input image and its corresponding segmentation. Similarly, the JEMs learn this between an input image and its corresponding augmented counterpart. Nevertheless, a notable difference is that the PU-Net does not share weights across both networks and that the model is inherently probabilistic. Given these additional complexities, we extend the PU-Net with projection layers on both the prior and posterior densities.

## 2.3 Generative modeling with Optimal Transport

Bousquet et al. (2017) proposed building a generative model from the perspective of minimizing the Wasserstein OT plan between the model and target distribution. Tolstikhin et al. (2019) built a generative model, similar to that of a VAE, with a Wasserstein objective and show that it can generate samples of higher quality. Patrini et al. (2020) have extended the analysis of Bousquet et al. (2017) and introduced a training objective that uses Sinkhorn iterations (Sinkhorn & Knopp (1967); Cuturi (2013b); Feydy et al. (2019)) to approximate the Wasserstein distance. We also exploit the Sinkhorn iterations for similar purposes. A detailed explanation can be found in Section 3.3.

# 3 Theory

We use calligraphic letters $(\mathcal{X})$ to denote sets, capital letters (X) for random variables and lower case letters $x$ for specific values. We denote marginals as $P_\mathrm{X}$, probability distributions as $P(\mathrm{X})$ and densities as $p(x)$. We specifically distinguish vectors and matrices with boldface characters. The fundamentals of Normalizing Flows are explained in Section 3.1. We discuss the Variational Autoencoder in Section 3.2 and the Probabilistic U-net in Section 3.2.1. We find compelling evidence that singular values can vanish in models such as the PU-Net. The Optimal Transport problem is formulated in Section 3.3. Furthermore, it is shown that conditional latent variable architectures are suitable to be used for OT problems and the choice of the Wasserstein distance as a divergence measure is justified in Section 3.3.1. Finally, an approximation to the Wasserstein distance, the Sinkhorn Algorithm, is discussed in Section 3.3.2.

## 3.1 Normalizing Flows

Normalizing Flows are a sequence of bijective transformations, typically starting from a complex distribution, transforming into a Normal distribution. Consider the variables $\mathbf{x}, \mathbf{z} \in \mathbb{R}^D$, where $\mathbf{z_0}$ is sampled from a Normal density $p_0(\mathbf{z})$ and $\mathbf{x}$ is the result of $\mathbf{z}_0$ subject to an NF with transformation $f_i : \mathbb{R}^D \mapsto \mathbb{R}^D$. The log-likelihood of the data, $\log p(\mathbf{x})$, can be stated as

$$\log p(\mathbf{x}) = \log p_0\left(\mathbf{z}_0\right) - \sum_{i=1}^{K} \log\left(\left|\det \frac{df_i}{d\mathbf{z}_{i-1}}\right|\right), \tag{1}$$

where the latent sample $\mathbf{z}_i$ is from the $i$-th transformation in the $K$-step NF. A planar flow, introduced by Rezende & Mohamed (2015), is a specific type of bijection that possesses the property to expand and contract distributions along a specific direction with the following transformation

$$f(\mathbf{z}) = \mathbf{z} + \mathbf{u}h(\mathbf{w}^T\mathbf{z} + b), \tag{2}$$

with $h$ being any smooth element-wise non-linear function, parameters $\mathbf{w} \in \mathbb{R}^D$, $\mathbf{u} \in \mathbb{R}^D$ and $b \in \mathbb{R}$ and the requirement $\mathbf{w}^T\mathbf{u} > -1$ .

## 3.2 Variational Autoencoders

Let $\mathbf{y} \in \mathcal{Y}$ be a observable variable taking values in $\mathbb{R}^D$. We define $\mathbf{z} \in \mathcal{Z}$ as a latent distribution of a lower-dimensional representation of $\mathbf{y}$, taking values in $\mathbb{R}^d$. It is assumed that $\mathcal{Y}$ possess a low-dimensional structure in $\mathbb{R}^r$ relative to the high-dimensional ambient space $\mathbb{R}^D$. We denote the probability measure on $\mathcal{Y}$, as $\mu$, such that the probability mass of an infinitesimal $d\mathbf{y}$ is $\mu(d\mathbf{y})$ and $\int_{\mathcal{Y}} \mu(d\mathbf{y}) = 1$. Furthermore, we assume $D \gg r$, i.e., the dimensionality of the observed variable is much greater than the embedded manifold. The variational autoencoder of Kingma & Welling (2014) trains a density model which attempts to learn

$p_Y(\mathbf{y})$ by optimizing the Evidence Lower Bound (ELBO) via an encoder-decoder structure as

$$p_Y(\mathbf{y}) = \int_{\mathcal{Y}} \left\{ \mathbb{E}_{q_\theta(\mathbf{z}|\mathbf{y})} \left[ \log \frac{p(\mathbf{y}, \mathbf{z})}{q_\theta(\mathbf{z}|\mathbf{y})} \right] + \text{KL} \left[ q_\theta(\mathbf{z}|\mathbf{y}) \,||\, p(\mathbf{z}|\mathbf{y}) \right] \right\} \mu(d\mathbf{y}) \tag{3a}$$

$$\geq \int_{\mathcal{Y}} \left\{ \mathbb{E}_{q_\theta(\mathbf{z}|\mathbf{y})} \left[ \log \frac{p(\mathbf{y}, \mathbf{z})}{q_\theta(\mathbf{z}|\mathbf{y})} \right] \right\} \mu(d\mathbf{y}) \tag{3b}$$

$$\geq \int_{\mathcal{Y}} \left\{ \mathbb{E}_{q_\theta(\mathbf{z}|\mathbf{y})} \left[ \log \frac{p_\phi(\mathbf{y}|\mathbf{z}) p(\mathbf{z})}{q_\theta(\mathbf{z}|\mathbf{y})} \right] \right\} \mu(d\mathbf{y}) \tag{3c}$$

$$\geq \int_{\mathcal{Y}} \left\{ \mathbb{E}_{q_\theta(\mathbf{z}|\mathbf{y})} \left[ \log p_\phi(\mathbf{y}|\mathbf{z}) \right] - \text{KL} \left[ q_\theta(\mathbf{z}|\mathbf{y}) \,||\, p_Z(\mathbf{z}) \right] \right\} \mu(d\mathbf{y}) = \text{ELBO}, \tag{3d}$$

with fixed prior $p_Z(\mathbf{z}) = \mathcal{N}(0, \mathbf{I})$, and where encoder $p_\theta(\mathbf{z}|\mathbf{y})$ and decoder $p_\psi(\mathbf{y}|\mathbf{z})$ are tractable distributions. In most cases, these are approximated by axis-aligned Normal densities modeled by arbitrary complex functions (e.g. a neural network). This implies $p_\theta(\mathbf{z}|\mathbf{y}) = \mathcal{N}(\boldsymbol{\mu}_\theta(\mathbf{y}), \boldsymbol{\Sigma}_\theta(\mathbf{y}))$ and $p_\psi(\mathbf{y}|\mathbf{z}) = \mathcal{N}(\boldsymbol{\mu}_\psi(\mathbf{z}), \gamma \cdot \mathbf{I})$. In most practical applications, $\gamma$ is set to unity. This design choice can result in blurry reconstructions. However, when $\gamma$ is learned, the reconstruction cost of the VAE objective favors small $\gamma \to 0$ (Dai & Wipf (2019)). By further inspection of the KL divergence term, Dai & Wipf (2019) show that this causes the VAE to naturally minimize the number of latent dimensions when $d > r$. The relevant latent dimensions will converge to low-variance noise, while additional redundant dimensions will match the prior distribution variance. This has detrimental effects when averaging across the training set, where it can be observed that

$$q_\theta(\mathbf{z}) = \int_{\mathcal{Y}} q_\theta(\mathbf{z}|\mathbf{y}) \mu(d\mathbf{y}) \not\approx \int_{\mathbb{R}^D} p_\phi(\mathbf{y}|\mathbf{z}) p_Z(\mathbf{z}) d\mathbf{y} = p_Z(\mathbf{z}). \tag{4}$$

In other words, the aggregated latent posterior is ill-matched to that of the prior. Generating images from the model density $p_{\hat{Y}}$ can be done by sampling from $p_Z$ and running the latent sample through the decoder as

$$p_{\hat{Y}}^{VAE}(\hat{\mathbf{y}}) := \int_{\mathbb{R}^d} p_\phi(\hat{\mathbf{y}}|\mathbf{z}) \, p_Z(\mathbf{z}) d\mathbf{z}. \tag{5}$$

Given an auxiliary input $\mathbf{x} \in \mathcal{X}$ taking values in $\mathbb{R}^D$ and with probability measure $\nu$, both the encoder and decoder of the conditional VAE (cVAE) are conditioned on $\mathbf{x}$. Thus, the generating process can be defined as

$$p_{\hat{Y}}^{cVAE}(\hat{\mathbf{y}}) := \int_{\mathbb{R}^D} \int_{\mathbb{R}^d} p_\phi(\hat{\mathbf{y}}|\mathbf{z}, \mathbf{x}) \, p_Z(\mathbf{z}) d\mathbf{z} d\mathbf{x}. \tag{6}$$

### 3.2.1 The Probabilistic U-Net

The Probabilistic U-Net (Kohl et al. (2018)) attempts to learn the variability in the mutual information between the input and ground-truth data with a cVAE-like structure. In contrast to a traditional cVAE with fixed prior, the PU-Net learns a prior to obtain image-dependent latent spaces. The ELBO is in this case

$$p_Y(\mathbf{y}) \geq \int_{\mathcal{Y}} \int_{\mathcal{X}} \left\{ \mathbb{E}_{q_\theta(\mathbf{z}|\mathbf{y}, \mathbf{x})} \left[ \log p_\phi(\mathbf{y}|\mathbf{z}, \mathbf{x}) \right] - \text{KL} \left[ p_\theta(\mathbf{z}|\mathbf{x}, \mathbf{y}) \,||\, p_\psi(\mathbf{z}|\mathbf{x}) \right] \right\} \nu(d\mathbf{x}) \mu(d\mathbf{y}), \tag{7}$$

where $p_\psi(\mathbf{z}|\mathbf{x}) = \mathcal{N}(\boldsymbol{\mu}_\psi(\mathbf{x}), \boldsymbol{\Sigma}_\psi(\mathbf{x}))$. Thus, instead of learning the data distribution, the PU-Net learns the distribution over a single datapoint. The decoder is dependent on $\mathbf{x}$, and the latent variable $\mathbf{z}$ is only injected in the final stages of the decoding U-Net. As an effect, the PU-Net encoder is not encouraged to learn a lower-dimensional representation of the ground-truth, but rather only the variability of it. Hence, the latent space dimensionality can be chosen to be much smaller than that of a conventional VAE. Since the prior is learned, its density is able to match to that of the posterior network, which has shown to learn low variances around optimal solutions. This can be problematic for generalization of the prior density as it can result in an under-utilized latent space, which limits its ability to fully encapsulate the aleatoric uncertainty present in the data. Our experiments indicate that some dimensions in the prior density collapse to low variances, even after optimizing the latent dimensionality, which in turn results in sub-optimal performance.

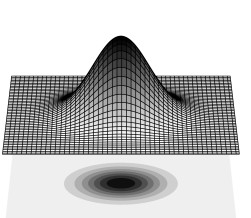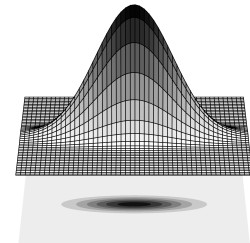

Figure 1: Normal densities with retained singular values (left, $\sigma_1 \approx \sigma_2$) and a vanished singular value (right, $\sigma_1 \gg \sigma_2$ or $\sigma_2 \gg \sigma_1$).

### 3.2.2 Latent singular values

Considering the fact that the modeled Normal densities are axis-aligned, the covariance matrices are diagonal and their entries represent its singular values, such that there is no need for singular value decomposition. Given $\boldsymbol{\sigma} = [\sigma_1, \sigma_2, ...\sigma_d]$, we can denote $\boldsymbol{\Sigma} = \text{diag}(\boldsymbol{\sigma})$. For simplicity, suppose $d = 2$, if the singular values are vanishing, there are significant difference in their relative magnitudes indicating $\sigma_1 \gg \sigma_2$ or $\sigma_2 \gg \sigma_1$. The opposite holds true when $\sigma_1 \approx \sigma_2$. Both cases are visualized in Figure 1.

### 3.3 Optimal Transport

Let $\mathcal{Y}$ and $\hat{\mathcal{Y}}$ be the two separable metric spaces. We adopt the Monge-Kantorovich formulation of Villani (2008) for the OT problem, by specifying

$$W^*(\mu, \hat{\mu}) := \inf \left\{ \iint_{\mathcal{Y} \times \hat{\mathcal{Y}}} c(\mathbf{y}, \hat{\mathbf{y}}) \, d\gamma(\mathbf{y}, \hat{\mathbf{y}}) \middle| \gamma \in \Gamma(\mu, \hat{\mu}) \right\}, \tag{8}$$

where $\Gamma(\mu, \hat{\mu})$ denotes the tight collection of all probability measures on $\mathcal{Y} \times \hat{\mathcal{Y}}$ with marginals $\mu$ and $\nu$, respectively, coupling $\gamma$, and $c(\mathbf{y}, \hat{\mathbf{y}}) : \mathcal{Y} \times \hat{\mathcal{Y}} \to \mathbb{R}_+$ being any lower semi-continuous measurable cost function. The usual context of this formulation is in finding the lowest cost of moving samples from the probability measures in $\mathcal{Y}$ to the measures in $\hat{\mathcal{Y}}$. In the case of probabilistic segmentation, the aim is to learn the ground-truth distribution $P_Y$ by matching it with the prediction distribution $P_{\hat{Y}}$.

### 3.3.1 Latent variable optimization

We consider the previously defined variables $\mathbf{y} \in \mathcal{Y}$ and $\mathbf{x} \in \mathcal{X}$ defined on $\mathbb{R}^D$, and the latent variable $\mathbf{z} \in \mathcal{Z}$ on $\mathbb{R}^d$. The decoder is chosen to be a deterministic function G with density $p_G(\hat{\mathbf{y}}|\mathbf{z}, \mathbf{x})$. We define our generative conditional latent model as

$$p_{\hat{Y}|X}(\hat{\mathbf{y}}|\mathbf{x}) := \int_{\mathbb{R}^D} \int_{\mathbb{R}^d} p_G(\hat{\mathbf{y}}|\mathbf{z}, \mathbf{x}) p_\theta(\mathbf{z}|\mathbf{x}) d\mathbf{z} d\mathbf{x}, \tag{9}$$

with $p_\theta(\mathbf{z}|\mathbf{x})$ as defined in Section 3.2 for the PU-Net. Furthermore, $\hat{Y}$ is a prediction defined on set $\hat{\mathcal{Y}}$ defined on $\mathbb{R}^D$, sampled from the model distribution $P_{\hat{Y}}$ with a density given by $p_{\hat{Y}}(\hat{\mathbf{y}}) = \int_{\mathbb{R}^D} p_{\hat{Y}|X}(\hat{\mathbf{y}}|\mathbf{x}) d\mathbf{x}$. Let $(Y, X) \sim P_{Y,X}$ be an input-target pair with induced marginals $P_Y$ and $P_X$. We define the prior encoder $P(Z|X)$ on set of prior distributions $\mathcal{P}$, with density $p_Z(\mathbf{z}) = \int_{\mathbb{R}^D} p_{Z|X}(\mathbf{z}|\mathbf{x}) d\mathbf{x}$ and marginal $P_Z$. Next, we consider the posterior encoder $Q(Z|X, Y)$ defined on the set of posterior distributions $\mathcal{Q}$ with density $q_Z(\mathbf{z}) = \int_{\mathbb{R}^D} \int_{\mathbb{R}^D} p_{Z|X,Y}(\mathbf{z}|\mathbf{x}, \mathbf{y}) d\mathbf{x} d\mathbf{y}$ and marginal $Q_Z$. We can rephrase the first theorem in the work of Bousquet et al. (2017) as follows.

**Theorem 1** *Let $G : \mathcal{Z} \times \mathcal{X} \to \hat{\mathcal{Y}}$ be purely deterministic, then*

$$W^*(P_Y, P_{\hat{Y}}) = \inf_{Q \in \mathcal{Q}: Q_Z = P_Z} \mathbb{E}_{X,Y \sim P_{X,Y}} \mathbb{E}_{Z \sim Q}[c(Y, G(Z, X))]. \tag{10}$$

Notice that the decoder is dependent on both the latent variable Z as well as the input image X. As proposed by Bousquet et al. (2017), the equality condition on $Q_Z$ and $P_Z$ can be relaxed by constraining the latent densities as stated in the second theorem below.

**Theorem 2** *Given a convex penalty $F : \mathcal{Q} \times \mathcal{P} \to \mathbb{R}_+$ such that $F(Q_Z, P_Z) = 0$ if and only if $Q_Z = P_Z$ and $\lambda > 0$, then*

$$W^\lambda(P_Y, P_{\hat{Y}}) = \inf_{Q \in \mathcal{Q}} \mathbb{E}_{X,Y \sim P_{X,Y}} \mathbb{E}_{Z \sim Q}[c(Y, G(Z, X))] + \lambda \cdot F(Q_Z, P_Z) \le W(P_Y, P_{\hat{Y}}), \tag{11}$$

*where the left-hand side approaches the upper bound with increasing $\lambda$.*

Hence, $W_c^\lambda(P_Y, P_{\hat{Y}})$ is a lower bound to $W_c(P_Y, P_{\hat{Y}})$ and minimization of the former does not guarantee minimization of the latter objective. Nevertheless, Tolstikhin et al. (2017) show that the objective improves upon the VAE with specific choices for $F$. Patrini et al. (2020) use in the penalty $F$ the $p$-Wasserstein distance, which is defined by

$$W_p(Q_Z, P_Z) := \inf \left\{ \left( \int_{\mathcal{Q} \times \mathcal{P}} d(\mathbf{q}, \mathbf{p})^p \, d\gamma(\mathbf{q}, \mathbf{p}) \right)^{\frac{1}{p}} \middle| \gamma \in \Gamma(Q_Z, P_Z) \right\}. \tag{12}$$

Here, the cost function is a distance metric $d$. According to Patrini et al. (2020), this choice of $F$ in Equation (2) makes $W^\lambda(P_Y, P_{\hat{Y}})$ an upper bound on $W_p(P_Y, P_{\hat{Y}})$. Thus, we rephrase Theorem 2.1 by Patrini et al. (2020) as follows.

**Theorem 3** *Let $p \ge 1$ and $G : \mathcal{Z} \times \mathcal{X} \to \hat{\mathcal{Y}}$ be any deterministic function that is $\gamma$-Lipschitz, then we obtain the equality*

$$W_p(P_Y, P_{\hat{Y}}) = \inf_{Q \in \mathcal{Q}} \sqrt[p]{\mathbb{E}_{X,Y \sim P_{X,Y}} \mathbb{E}_{Z \sim Q}[c(Y, G(Z, X))^p]} + \gamma \cdot W_p(Q_Z, P_Z), \tag{13}$$

*where $\mathcal{Q}$ is any class of probabilistic encoders that at least contains a class of universal approximators.*

This objective is resembles the ELBO, which is similarly composed of a reconstruction and divergence term. This theorem justifies the use of the Wasserstein distance as a divergence measure in our conditional probabilistic segmentation model. Note that in contrast to previous works, we allow the network to learn the appropriate input-conditional prior density.

### 3.3.2 Sinkhorn Algorithm as a Wasserstein approximation

In practice, the intricacies of the Wasserstein distance complicate its calculation. It has been proposed by Wilson (1968) to introduce entropic regularization to the OT problem. This is achieved by using the entropy of the couplings as a regularizing function, which is specified by

$$\tilde{S}_\epsilon(\mu, \hat{\mu}) := \inf \left\{ \int_{\mathcal{Y} \times \hat{\mathcal{Y}}} \left( d(\mathbf{y}, \hat{\mathbf{y}}) + \epsilon \log \frac{d\gamma(\mathbf{y}, \hat{\mathbf{y}})}{d\mu(\mathbf{y}) d\hat{\mu}(\hat{\mathbf{y}})} \right) d\gamma(\mathbf{y}, \hat{\mathbf{y}}) \middle| \gamma \in \Gamma(\mu, \hat{\mu}) \right\}, \tag{14}$$

where $\epsilon \in \mathbb{R}_+$. As mentioned by Cuturi (2013a), the entropy term can be expanded to $\log(d\gamma(\mathbf{y}, \hat{\mathbf{y}})) - \log(d\mu(\mathbf{y})) - \log(d\hat{\mu}(\hat{\mathbf{y}}))$. In this way, this formulation can be understood as constraining the joint probability to have *sufficient entropy* or contain small enough *mutual information* with respect to $d\mu$ and $d\hat{\mu}$. This entropic regularization allows optimization over a Lagrangian dual for faster computation with the iterative Sinkhorn matrix scaling algorithm. The entropic bias is removed from the OT problem to obtain the Sinkhorn Divergence, specified as

$$S_\epsilon(\mu, \nu) = \tilde{S}_\epsilon(\mu, \nu) - \frac{1}{2} \left( \tilde{S}_\epsilon(\mu, \mu) + \tilde{S}_\epsilon(\nu, \nu) \right). \tag{15}$$

The Sinkhorn divergence interpolates between $W_p$ ($\epsilon \to 0$) with $\mathcal{O}(\epsilon \log(\frac{1}{\epsilon}))$ deviation and Maximum Mean Discrepancy ($\epsilon \to \infty$), which favours dimension-independent sample complexity (Genevay et al. (2019)). A viable option is to approximate the Sinkhorn Divergence via sampling with weights $\alpha, \beta \in \mathbb{R}_+$. The regularized Sinkhorn algorithm performance lacks in lower temperature settings. To alleviate this limitation, as well as to increase efficiency, Kosowsky & Yuille (1994) introduce $\epsilon$-*scaling* or *simulated annealing* to the Sinkhorn algorithm.

# 4    Methods

In the context of density modeling, several criteria need to be satisfied in order to employ latent space projection. Following previous works, we employ projection layers during training but only use the preceding embeddings after training. In other words, we remove the NFs after training and sample directly from the axis-aligned Normal densities. Firstly, we constrain the projected latent spaces during training. For Normal densities, the KL-divergence can be used. However, when the two underlying distributions are unknown (i.e. only a set of samples are observed), the KL-divergence is not a viable objective. This can be seen by substituting Equation (1) in Equation (7) for both prior and posterior NFs. However, we can use the Sinkhorn Divergence for an approximate Wasserstein distance and employ this as a latent space constraint (Theorem 3). To achieve this, the sample likelihood needs to remain known after the projecting function. NFs are a suitable transformation due to their bijective property that allows keeping track of the sample probabilities with Equation (1). Finally, the preceding latent space needs to be sufficiently close to the projected latent space, such that samples result in accurate segmentation reconstructions.

In the cSAE, the projections originate from the base Normal distributions which can theoretically be unconstrained. Nevertheless, we empirically found that this leads to overfitting. As a solution to overfitting, we additionally penalize the KL-divergence between the base Normal distributions. This has several merits. Firstly, this constraint forces the posterior network to learn the mutual information between the input and ground-truth images in the Normal densities. Without the constraint, a significant part of information is attempted to be discarded in the augmented NFs, which are notoriously inexpressive. Furthermore, this design allows for the NFs to eventually converge to identical densities. In other words, initially, during training the projected space is far from the base Normal densities. As training continues, the projected latent spaces coincide with the base Normal densities. This enables us to remove the augmented NFs and sample accurate predictions from the preceding base Normal distributions. We hypothesize that this method results in better latent densities with retained singular values, due to the regularizing effect of the projected space.

Combining the aforementioned approaches, we can implement the conditional Sinkhorn Autoencoder, as presented in Figure 2. The posterior network $Q$, which attempts to learn the ambiguity in the image, is conditioned on the input image X and ground-truth labels Y. The prior and posterior networks $P$ and $Q$ output Normal densities $\mathcal{N}_P$ and $\mathcal{N}_Q$, as well as the marginals induced by the NFs, which we refer to as $Q_{\mathrm{Z}}$ and $P_{\mathrm{Z}}$, respectively. During training, samples are drawn from the posterior and combined with the features from the generator (in our case the U-Net of Ronneberger et al. (2015)), to construct a segmentation. We state the training objective of the cSAE as

$$\mathcal{L} = -\mathbb{E}_{Z \sim Q}\mathbb{E}_{X \sim P_{\mathrm{X}}}[\log p_{\mathrm{G}}(Y|Z, X)] + \beta \cdot \mathrm{KL}\left(\mathcal{N}_Q || \mathcal{N}_P\right) + \gamma \cdot S_\epsilon(Q_{\mathrm{Z}}, P_{\mathrm{Z}}), \qquad (16)$$

where $\beta$ and $\gamma$ are tuneable hyperparameters introduced to regulate influence of the KL and Sinkhorn Divergence.

## 4.1    Dataset details

Several post-processed multi-annotated public datasets are used for training. Firstly, we use the LIDC-IDRI (Armato III et al. (2011)) dataset containing CT scans of lung nodules with up to four annotations. Secondly, we use the QUBIQ 2021 (Menze et al. (2021)) dataset consists of 7 multi-annotated subsets containing CT and MRI imaging of the prostate, brain, kidney and pancreas. We have found that the PU-Net already generalizes well on the prostate and kidney datasets, leaving little room for improvements. Therefore, we have experimented with the remaining datasets. More details on the datasets and training procedure can be found in Appendices A and B, respectively.

## 4.2    Performance evaluation

To demonstrate the effectiveness of our approach, we compare the cSAE with the PU-Net and its NF-augmented posterior variant of Valiuddin et al. (2021). We keep the authors naming convention and refer to the posterior augmented model as 2$p$-planar, indicating a two-step planar NF on the posterior only. A

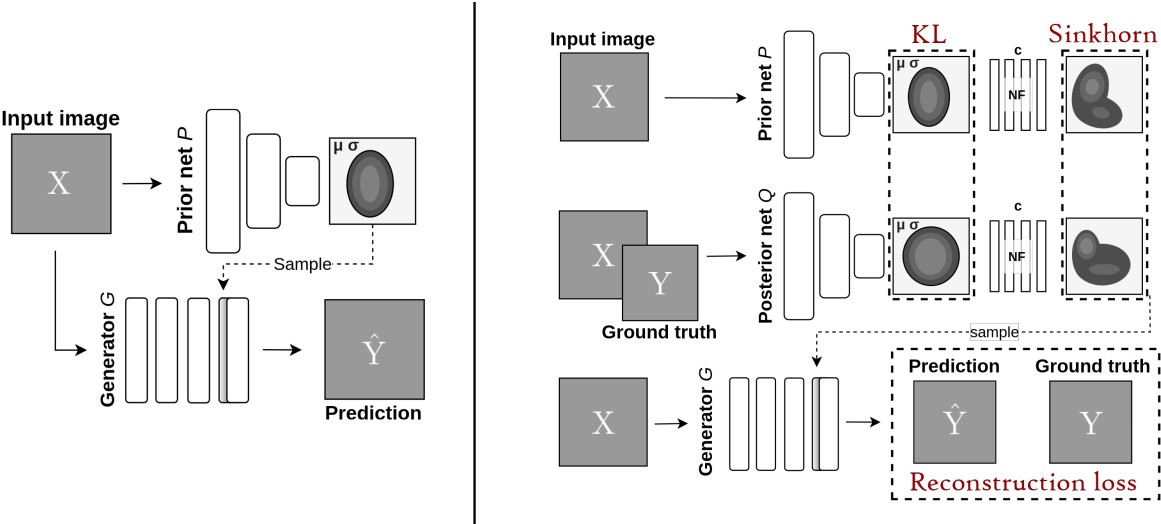

Figure 2: The Conditional Sinkhorn Autoencoder during testing (left) and training (right). Note how during training samples are taken from the projected, NF transformed latent space, and at test time this projection head is discarded.

two-layer non-linear projector head (i.e. a two-step NF) is used, since it has yielded substantial success (Jing et al. (2021); Valiuddin et al. (2021)).

We consider the inter-observer variability (i.e. disagreement between the annotators) of the multi-annotated datasets as the target for the learned aleatoric uncertainty. We use several evaluation metrics on our test sets to measure the similarity of the prediction and ground-truth images. Similarly to Kohl et al. (2019), Monte Carlo sampling (i.e. sampling from the latent prior for reconstruction) can be used to uniquely match all elements in the prediction with an element in the ground-truth set and is commonly referred to as Hungarian Matching. We refer to this method as the Empirical Wasserstein metric, denoted as $\widehat{W_k}$, because it can be regarded as a Monte Carlo approximation of the original Wasserstein objective defined in Equation (8). We calculate the Empirical Wasserstein metric by comparing a discrete set of predictions and ground-truth images via a coupling matrix $\tilde{\gamma}$. Each element $\tilde{\gamma}_{i,j}$ indicates the respective cost $k$ between $i$-th prediction and $j$-th ground-truth image. The optimal coupling $\tilde{\gamma}^*$ finds the minimal average cost $k$ of all unique matches in $\tilde{\gamma}$. To match the number of elements in both sets, we duplicate the number of ground-truth images to match the sample size of the predictions. When both the prediction and ground-truth image are empty, we assign maximum score (i.e. minimal distance). For the Empirical Wasserstein metric, four different kernels are used. We apply the Intersection over Union (IoU) defined as $\frac{TP}{TP+FP+FN}$, and the Dice score defined as $\frac{2 \cdot TP}{2 \cdot TP+FP+FN}$, where TP, FP, TN, FN stand for the pixel-wise true positive, false positive, true negative and false negative, respectively, between the ground-truth $\mathbf{y}$ and prediction images $\hat{\mathbf{y}}$. Furthermore, we make use of the Hausdorff distance. Let $\mathcal{C}$ (resp. $\hat{\mathcal{C}}$) be the set containing all non-zero pixel-coordinate vectors of ground-truth $\mathbf{y}$ (resp. prediction $\hat{\mathbf{y}}$), then the Hausdorff distance can be defined as

$$d_{HD}(\mathbf{y}, \hat{\mathbf{y}}) := \max\{\max_{\mathbf{c} \in \mathcal{C}} d(\mathbf{c}, \hat{\mathcal{C}}), \max_{\hat{\mathbf{c}} \in \hat{\mathcal{C}}} d(\mathcal{C}, \hat{\mathbf{c}})\}, \tag{17}$$

where $d(\mathbf{c}, \hat{\mathcal{C}}) = \min_{\hat{\mathbf{c}} \in \hat{\mathcal{C}}} d(\mathbf{c}, \hat{\mathbf{c}})$ and $d(\mathcal{C}, \hat{\mathbf{c}}) = \min_{\mathbf{c} \in \mathcal{C}} d(\mathbf{c}, \hat{\mathbf{c}})$, and $d$ denotes the Euclidean distance. Finally, we also evaluate the negative log-likelihood (cross-entropy) defined as

$$H_{ce}(\mathbf{y}, \hat{\mathbf{y}}) = -\sum_{i,j} y_{ij} \log(\hat{y}_{ij}), \tag{18}$$

Furthermore, to quantify the vanishing of the singular values we make use of the Effective Rank (ER) (Roy & Vetterli (2007)). This quantity is a real-valued extension of the matrix rank and can naturally quantify the sparsity of the covariance matrix $\mathbf{\Sigma}$. Because the Normal densities are axis-aligned, Singular Value

Decomposition is not required and we can directly obtain the $d$-dimensional singular value vector $\boldsymbol{\sigma}$ from the trace of $\boldsymbol{\Sigma}$. Then, singular values are normalized and ranked in an ascending order. The effective rank is defined as

$$\text{ER}(\boldsymbol{\sigma}) = e^{-\sum_k^d \sigma_k \log p_k} \tag{19}$$

Hurley & Rickard (2009) find that entropy-based metrics are not ideal measures of sparsity. Although the entropy-based metrics evaluated by Hurley & Rickard (2009) are not identical to the ER, we still additionally introduce the Gini index (Gini (1921); Lorenz (1905)), since it has shown to sufficiently adhere to their six criteria for an accurate sparsity measure. The Gini index can be formulated as

$$\text{Gini}(\boldsymbol{\sigma}) = 1 - 2 \sum_{k=1}^d \frac{\sigma_k}{\|\boldsymbol{\sigma}\|_1} \left( \frac{d - k + \frac{1}{2}}{d} \right). \tag{20}$$

All evaluations are done on a test set, for each of the cross-validation folds, to rule out convenient splitting of the dataset. To qualitatively depict the model performance, we present the mean and standard deviation of the sample reconstructions. Even though it can benefit the quality of the predictions, we refrain from using temperature scaling during inference as this provides a more accurate representation of the learning capability of the models.

## 5 Results and discussion

### 5.1 Quantitative evaluation

We present the test set evaluation of the three-fold cross-validation training in Table 1. Our main findings are twofold. Firstly, it is confirmed that by augmenting the PU-Net with NFs, the singular values are better retained. This can be understood from the Gini indices of the evaluations on the different datasets. In particular, it is observed that the cSAE has the lowest Gini index, indicating the least sparse singular values. The Gini indices of the 2$p$-planar models generally lie in between that of the PU-Net and the cSAE. This shows that only projecting the posterior can already assist in retaining the singular values. Yet, improvements introduced by the 2$p$-planar models are marginal on both retainment of the singular values and the other evaluation metrics. It can be seen that the best performance is obtained when projecting both densities with the cSAE. The low Gini indices and high ER values are a strong indication of retained singular values that enable diverse and plausible segmentations, captured by our other test metrics.

The latent space behaviour during training time is also investigated. We depict the Gini indices of the prior density singular values as the model converges in Figure 3. Interestingly enough, it can be observed that in most instances, the PU-Net prior latent space has singular values vanishing (increased Gini index) early in training. On the other hand, it can also be observed that the singular values of the 2$p$-planar and cSAE models always have a lower Gini index and are more volatile. We see a trend where more volatile Gini indices during training are also generally lower. We hypothesize that this volatility is due to the mechanism that retains the singular values during training time. We leave the details of this exact mechanism for future work. From the results it can be confirmed that the cSAE retains the singular values the most.

| Dataset | Subset | Model | $\widehat{W}_k(\mathcal{Y}, \hat{\mathcal{Y}}) \downarrow$ | | | | Gini↓ | ER↑ |
|---------|--------|-------|-----------|-----------|-----------|-----------|-------|-----|
| | | | 1 - IoU | 1 - Dice | $d_{HD}$ | $H_{ce}$ | | |
| LIDC-IDRI | | PU-Net | $0.363 \pm 0.049$ | $0.309 \pm 0.052$ | $21.426 \pm 4.616$ | $0.330 \pm 0.068$ | $0.708 \pm 0.067$ | $2.017 \pm 0.574$ |
| | | 2p-planar | $0.364 \pm 0.039$ | $0.309 \pm 0.037$ | $21.044 \pm 3.139$ | $0.329 \pm 0.054$ | $0.511 \pm 0.117$ | $3.504 \pm 0.941$ |
| | | cSAE | $\mathbf{0.346 \pm 0.038}$ | $\mathbf{0.292 \pm 0.038}$ | $\mathbf{19.896 \pm 3.161}$ | $\mathbf{0.303 \pm 0.052}$ | $\mathbf{0.351 \pm 0.031}$ | $\mathbf{4.754 \pm 0.131}$ |
| QUBIQ 2021 | pancreas | PU-Net | $0.332 \pm 0.092$ | $0.231 \pm 0.080$ | $11.699 \pm 5.915$ | $0.753 \pm 0.310$ | $0.756 \pm 0.014$ | $1.605 \pm 0.099$ |
| | | 2p-planar | $0.321 \pm 0.071$ | $0.221 \pm 0.059$ | $12.103 \pm 3.792$ | $0.692 \pm 0.222$ | $0.509 \pm 0.103$ | $3.354 \pm 0.766$ |
| | | cSAE | $\mathbf{0.286 \pm 0.080}$ | $\mathbf{0.193 \pm 0.067}$ | $\mathbf{9.804 \pm 4.854}$ | $\mathbf{0.550 \pm 0.194}$ | $\mathbf{0.362 \pm 0.102}$ | $\mathbf{4.750 \pm 0.616}$ |
| | pancreatic lesion | PU-Net | $0.323 \pm 0.078$ | $0.213 \pm 0.063$ | $9.045 \pm 2.760$ | $0.756 \pm 0.177$ | $0.786 \pm 0.020$ | $1.395 \pm 0.155$ |
| | | 2p-planar | $0.340 \pm 0.073$ | $0.232 \pm 0.065$ | $9.156 \pm 3.566$ | $0.738 \pm 0.207$ | $0.612 \pm 0.096$ | $2.745 \pm 0.833$ |
| | | cSAE | $\mathbf{0.290 \pm 0.075}$ | $\mathbf{0.186 \pm 0.063}$ | $\mathbf{7.150 \pm 3.124}$ | $\mathbf{0.634 \pm 0.230}$ | $\mathbf{0.406 \pm 0.080}$ | $\mathbf{4.350 \pm 0.556}$ |
| | brain growth | PU-Net | $0.349 \pm 0.003$ | $0.214 \pm 0.002$ | $6.728 \pm 0.185$ | $3.054 \pm 0.103$ | $0.278 \pm 0.127$ | $5.148 \pm 0.575$ |
| | | 2p-planar | $0.358 \pm 0.021$ | $0.220 \pm 0.016$ | $\mathbf{6.588 \pm 0.110}$ | $3.189 \pm 0.246$ | $0.372 \pm 0.111$ | $4.485 \pm 0.755$ |
| | | cSAE | $\mathbf{0.338 \pm 0.009}$ | $\mathbf{0.204 \pm 0.006}$ | $6.759 \pm 0.289$ | $\mathbf{2.960 \pm 0.102}$ | $\mathbf{0.218 \pm 0.066}$ | $\mathbf{5.500 \pm 0.232}$ |

Table 1: Comparison of the proposed cSAE with the PU-Net and its posterior NF-augmented variant (2p-planar) on various datasets. We present the mean and standard deviation (due to threefold cross-validation) of the Empirical Wasserstein metric for multiple kernels and the Gini index of the singular values. All results are test set evaluations. The cSAE has better performance on almost all metrics.

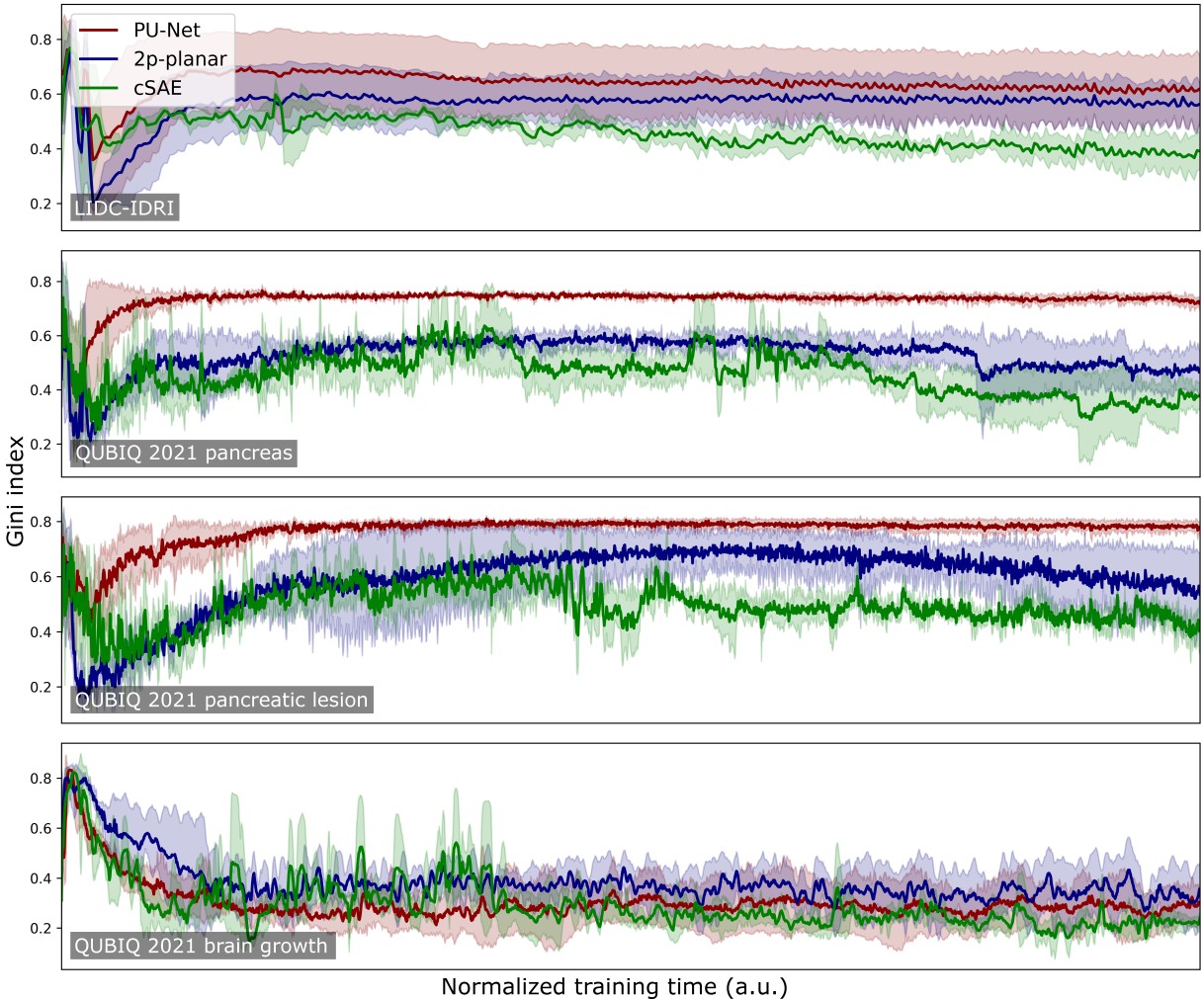

Figure 3: Gini indices of the prior singular values over normalized training time for the LIDC-IDRI and QUBIQ 2021 datasets. Lower is better. The cSAE has the lowest Gini index for all experiments, followed by the 2p-planar and PU-Net. This shows that latent space projections retain the density singular values.

## 5.2 Qualitative evaluation

Qualitative samples of the various dataset are shown in Appendix C. It is clearly visible that the cSAE better encapsulates the aleatoric uncertainty. For the LIDC dataset, it is observed that the PU-Net considers ambiguity present at the center of the lesions, while the cSAE accurately indicates most ambiguity almost exclusively around the edges. As a result of a better latent space, we observe stronger mean values in the areas where the annotators agree and stronger standard deviation values where the tend to annotators disagree. The cSAE aligns the closest with the inter-observer variability and continuously the aleatoric uncertainty in the data.

We also present an outlier case in Appendix D on the QUBIQ 2021 Brain tumor dataset, where the cSAE was not always the better performing model. We observe that the latent space with the least sparse singular value vector was not the cSAE nor was it the best performing model. These results can indicate that for this dataset and with the particular chosen latent space dimensionality (six), the sparsity of the singular values is not the bottleneck of the performance. It is noteworthy the differences in evaluations are marginal. This, combined with the fact that the values vary substantially per fold challenges the statistical significance of the results and reliability of any conclusions therefrom. We highlight two possible reasons for the deviating behaviour. Firstly, it is important to note that the Brain tumor subset is significantly smaller (only around 20 images), which is likely too low to sufficiently generalize on the problem. Secondly, it is the only multi-modal MRI dataset used in the experiments, where the input data consisted of different acquisition modalities, concatenated channel-wise.

## 6 Conclusion

Quantifying uncertainty in image segmentation is quintessential for well-informed decision making. In this paper, we propose to use Normalizing Flows and the theory of Optimal Transport to improve the latent modeling of the ambiguity in the data. Our approach, the cSAE, prevents vanishing of the singular values of modeled axis-aligned Normal densities, which is observed in the former variants of the Probabilistic U-Net. The cSAE produces accurate segmentation hypotheses as can be seen through extensive quantitative and qualitative evaluation. In our proposed framework, the absence of vanishing singular values results in significant improvements of the aleatoric uncertainty quantification. We propose that the projection of latent spaces with NFs and OT should be experimented with throughout other ambiguous modalities. Our future work will focus on gaining a deeper understanding on the exact mechanism that retains the singular values. Furthermore, we did not consider architectures that model densities at multiple resolutions. We are convinced that extending this work in the multi-resolution setting will similarly result in performance gains.

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

## A    Dataset details

The LIDC-IDRI dataset is preprocessed from 1,018 thoracic CT scans into 15,096 128×128-pixel patches. Each image has up to four annotators. The pancreas subset of the QUBIQ 2021 dataset contains 76 cases with 2 annotations per slice. We process the data such that each slice has at least one non-zero value in one of the associated annotations. This results in a total of 869 128×128-pixel patches. The brain-tumor subset has four input channels where each channel corresponds to a different acquisition method (multimodal MRI) for the same subject. For the other subsets, i.e. pancreatic lesion and brain growth, we only normalize and resize to 128×128-dimensional patches. The information on the datasets are summarized in Table 2

| Dataset | Datapoints | Dimensionality | Annotators | Input channels | Tasks |
|---|---|---|---|---|---|
| LIDC-IDRI | 15,096 | 128×128 | 4 | 1 | 1 |
| Pancreas | 869 | 128×128 | 2 | 1 | 1 |
| Pancreatic lesion | 156 | 128×128 | 2 | 1 | 1 |
| Brain growth | 39 | 128×128 | 7 | 1 | 1 |
| Brain tumor (all) | 32 | 128×128 | 7 | 4 | 4 |

Table 2: Details of each applied dataset.

## B    Training details

Training details across datasets are kept as similar as possible to rule out biases towards favourable test-set evaluations. We have summarized the hyperparameter settings per dataset in Table 3. The models are trained with the Adam optimizer, weight decay and early stopping applied to the validation loss. The hyperparameters $\beta$ and $\gamma$ weigh the contribution of the KL and Sinkhorn divergence towards the loss function. The KL divergence of the $L$-dimensional densities are calculated analytically and the Sinkhorn iterations are based upon $N_{OT}$ samples. All Monte-Carlo evaluations are done with $N_{eval}$ samples. Tuning of the number of iterations for early stopping is based upon the validation loss during training. For the aforementioned reason, we have also kept the data augmentation consistent throughout the datasets and models (see Table 4 for details). Splitting of the different datasets are also performed in the same manner. For testing, the first 20% of the dataset is used. The remaining 80% is used for training with threefold cross-validation. We have chosen cross-entropy for the reconstruction loss. For the Sinkhorn Divergence, the `GeomLoss` Feydy et al. (2019) library has been used with $p = 2$ and $Diameter = 100$. The other parameters for the Sinkhorn Divergence are left to default. The gradients are clipped to have unitary norm. Training has been done on an 11GB NVIDIA RTX 2080TI. Our implementation of the cSAE with KL regularization will be made public.

| Dataset | $\beta$ | $\gamma$ | Batch size | Learning rate | Weight decay | Patience | $L$ | $N_{OT}$ | $N_{eval}$ |
|---|---|---|---|---|---|---|---|---|---|
| LIDC-IDRI | 10 | 10 | 32 | $10^{-4}$ | $5e^{-5}$ | 40 | 6 | 16 | 16 |
| Pancreas | 10 | 1 | 32 | $10^{-4}$ | $5e^{-5}$ | 200 | 6 | 16 | 16 |
| Pancreatic lesion | 10 | 10 | 32 | $10^{-4}$ | $5e^{-5}$ | 500 | 6 | 16 | 15 |
| Brain growth | 10 | 10 | 32 | $10^{-4}$ | $5e^{-5}$ | 300 | 6 | 16 | 28 |
| Brain tumor (all) | 10 | 1 | 32 | $10^{-4}$ | $5e^{-5}$ | 300 | 6 | 16 | 15 |

Table 3: Training hyperparameters settings for the training of the different datasets.

| Random augmentation | Min | Max |
|---|---|---|
| Rotation | -180 | 180 |
| Translation | -0.1 | 0.1 |
| Scaling | 0.8 | 1.2 |
| Shear (x and y) | -30 | 30 |
| Brightness | 0.8 | 1.2 |
| Gamma | 0.5 | 1.5 |

Table 4: Data augmentation settings for all datasets.

## C  Qualitative samples

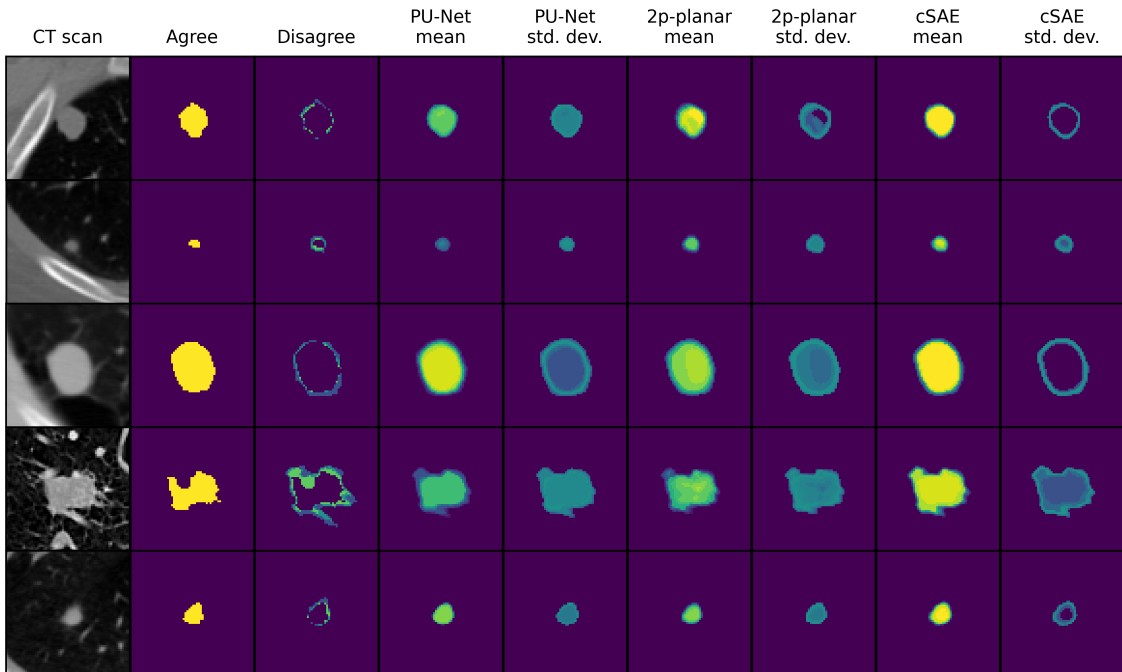

Figure 4: Test set samples of the LIDC IDRI dataset

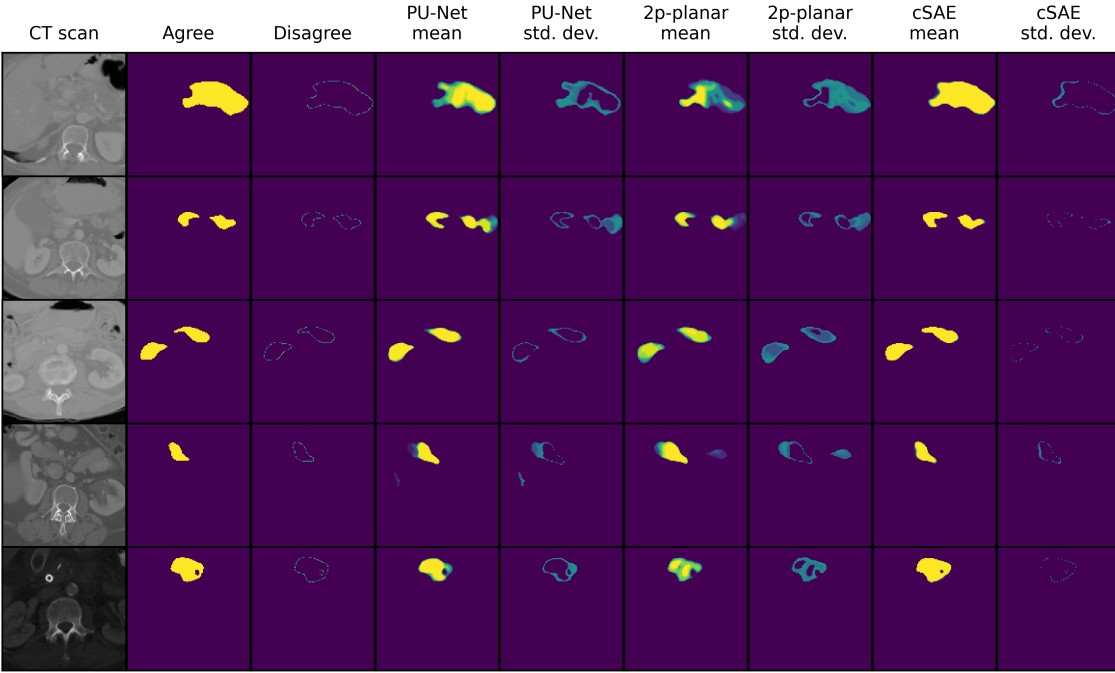

Figure 5: Test set samples of the QUBIQ 2021 pancreas subset

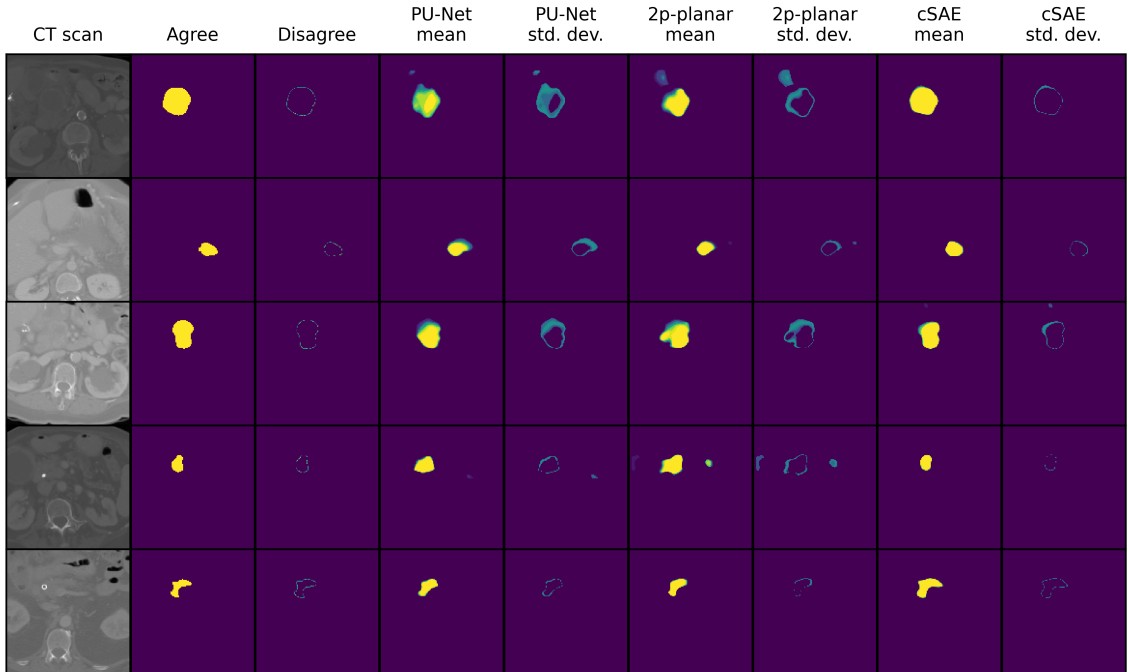

Figure 6: Test set samples of the QUBIQ 2021 pancreatic-lesion subset

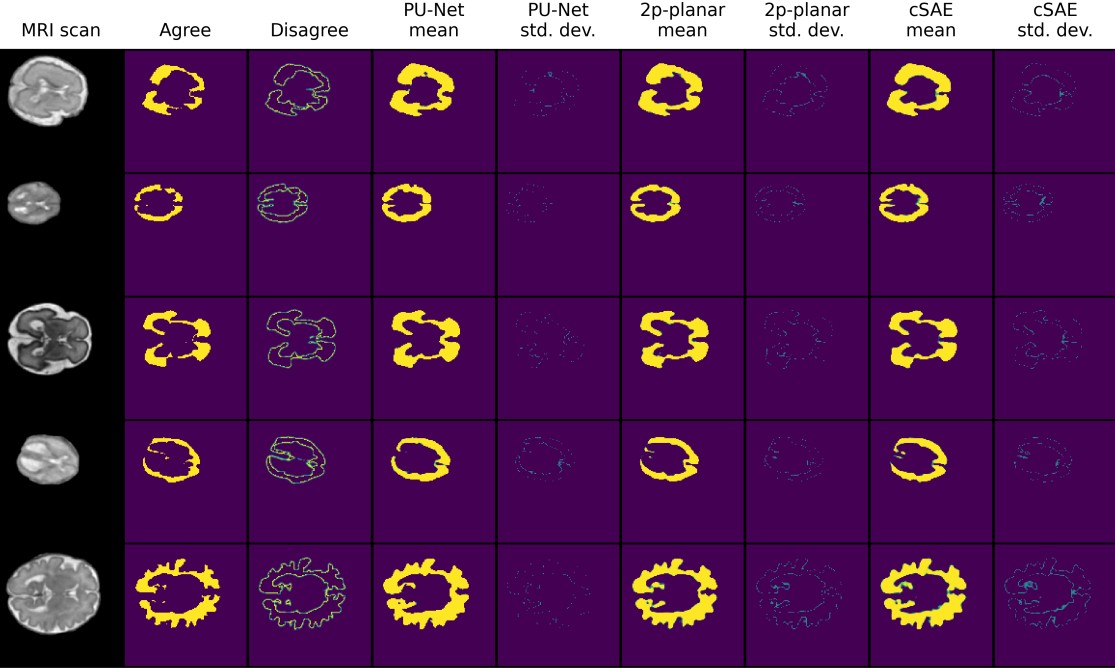

Figure 7: Test set samples of the QUBIQ 2021 brain-growth subset

# D    Outlier cases

We present the quantitative results on the QUBIQ 2021 Brain tumor dataset in Table 5 and Figure 8.

| Dataset | Subset | Model | $\widehat{W}_k(\mathcal{Y}, \hat{\mathcal{Y}}) \downarrow$ | | | | **Gini↓** | **ER↑** |
| | | | 1 - IoU | 1 - Dice | Hausdorff | NLL | | |
|---|---|---|---|---|---|---|---|---|
| QUBIQ 2021 brain tumor | task 1 | PU-Net | $0.619 \pm 0.022$ | $0.475 \pm 0.021$ | $19.591 \pm 3.294$ | $2.167 \pm 0.164$ | $0.363 \pm 0.132$ | $4.636 \pm 0.773$ |
| | | 2$p$-planar | $\mathbf{0.603 \pm 0.007}$ | $\mathbf{0.460 \pm 0.013}$ | $\mathbf{17.291 \pm 2.969}$ | $\mathbf{2.127 \pm 0.139}$ | $\mathbf{0.204 \pm 0.031}$ | $\mathbf{5.541 \pm 0.084}$ |
| | | cSAE | $0.625 \pm 0.018$ | $0.485 \pm 0.018$ | $19.188 \pm 2.808$ | $2.432 \pm 0.270$ | $0.318 \pm 0.136$ | $4.914 \pm 0.787$ |
| | task 2 | PU-Net | $0.481 \pm 0.084$ | $0.377 \pm 0.085$ | $18.505 \pm 5.574$ | $1.048 \pm 0.284$ | $0.712 \pm 0.146$ | $2.019 \pm 1.233$ |
| | | 2$p$-planar | $\mathbf{0.412 \pm 0.107}$ | $\mathbf{0.309 \pm 0.102}$ | $\mathbf{13.016 \pm 7.982}$ | $\mathbf{0.835 \pm 0.286}$ | $0.622 \pm 0.132$ | $2.769 \pm 1.116$ |
| | | cSAE | $0.431 \pm 0.096$ | $0.313 \pm 0.080$ | $16.599 \pm 10.176$ | $1.532 \pm 0.719$ | $\mathbf{0.610 \pm 0.147}$ | $\mathbf{2.906 \pm 1.217}$ |
| | task 3 | PU-Net | $0.651 \pm 0.011$ | $0.579 \pm 0.011$ | $17.054 \pm 0.241$ | $\mathbf{1.280 \pm 0.010}$ | $0.400 \pm 0.076$ | $4.515 \pm 0.469$ |
| | | 2$p$-planar | $\mathbf{0.641 \pm 0.011}$ | $\mathbf{0.568 \pm 0.010}$ | $\mathbf{16.638 \pm 0.152}$ | $1.291 \pm 0.016$ | $\mathbf{0.215 \pm 0.083}$ | $\mathbf{5.491 \pm 0.378}$ |
| | | cSAE | $0.647 \pm 0.012$ | $0.574 \pm 0.010$ | $16.785 \pm 0.082$ | $1.285 \pm 0.006$ | $0.397 \pm 0.222$ | $4.265 \pm 1.303$ |

Table 5: Comparison of the proposed cSAE with the PU-Net and its posterior NF-augmented variant (2$p$-planar) on the Brain tumor subset of the QUBIQ 2021 dataset. We present the mean and standard deviations (due to threefold cross-validation) of the Empirical Wasserstein metric for multiple kernels and the Gini indices of the singular values. All results are test set evaluations.

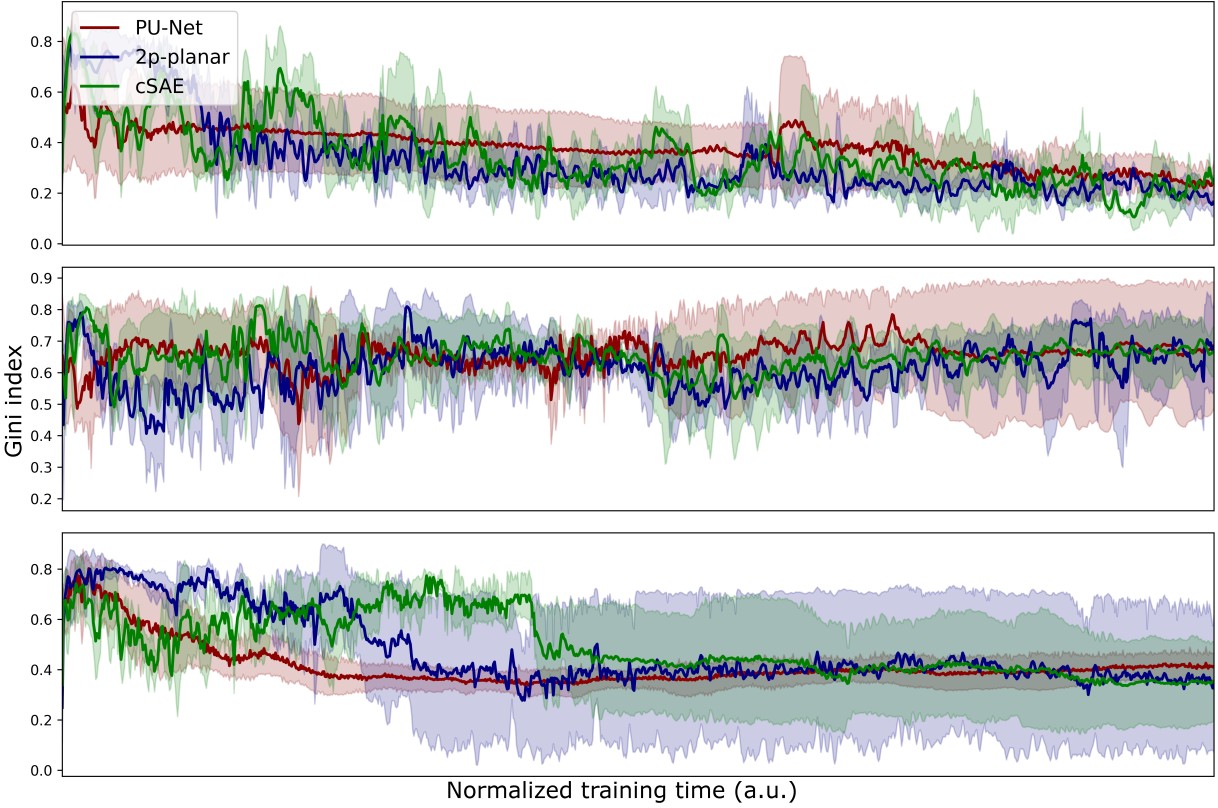

Figure 8: Gini indices of the prior singular values over normalized training time for the three tasks of the QUBIQ 2021 Brain tumor dataset. Lower is better. The various models do not portray a clear relationship with the Gini index, since their respective values are close and have high variance.

