# OpenReview forum: "Retained Singular Values in Probabilistic Image Segmentation with Normalizing Flows and Optimal Transport"
_TMLR — Rejected by TMLR_

### Review · Reviewer_coKj · 2022-07-16

**Summary Of Contributions:**

The paper claims they propose a new objective to alleviate the 'vanished singular value' problem. However, since the paper is not well presented, so it is not clear the problem they want to solve, what the proposed method is and why it can help to solve the problem.

**Broader Impact Concerns:**

I didn't realize any broader impact since I couldn't understand the paper.

**Requested Changes:**

I recommend a strong rejection of the paper.
I think the author should rewrite the paper by making the following points clear to make it a valid paper before sending it to review.

1. Formally describe your task and give an introduction to the conditional VAE method.
2. Describe the singular value vanishing problem in the introduced task and models.
3. Give intuition about how to alleviate the problem.
4. Use the right math notation to describe the proposed method and why it can help alleviate the problem.






**Strengths And Weaknesses:**

## Weakness
The paper is poorly written and is not ready to be a qualified scientific paper at a basic level.
### The motivation is not clear.
The authors mentioned the vanished singular. However, there is no clear description of the problem but only Figure  1 with no explanations, that try to illustrate the problem.  The paper cites some self-supervised learning papers, but I don't know why they relate to the target task in this paper, the models and goals are completely different.

### The methodology is not clear.
Both Figure 2 and equation 12 are not clear. For example, the paper denotes Q as a posterior network, so what does $Z\sim Q$ means? Or  "the sample after NF" as $\hat{Q}$ or $\hat{P}$, what does it mean? Also, what is the role of normalizing flow here?
Figure 2 shows the distribution is based on one data, but in section 3.2.1 they define the Wasserstein distance between aggregation posterior and prior.

I also cannot comment on the correctness or the novelty of the method since the description of the method is not scientific and hard to understand.

Others:
1. The planar flow mentioned in section 3.1 is not bijective without further constrain.
2. Section 3.2  looks like a sloppy updation from other sources, it is hard to believe that section 3.2 and section 4 are written by the same person.

---

> ### Author Response · Authors · 2022-09-26
> **Reply to reviewer coKj**
>
> The authors would like to thank the reviewer for the comments and feedback. Since we are building on prior work and wanted to keep the paper concise, we likely did not provide sufficient background information on the topic and did not appropriately set the scene for the novelty we propose. Regarding the weaknesses:
>
> **The authors mentioned the vanished singular. However, there is no clear description of the problem but only Figure 1 with no explanations, that try to illustrate the problem.**
>
> We have updated our paper with an additional Section (3.4), which goes in further detail on the vanishing singular values. We also moved the reference to Figure 1 to this section to avoid confusing the reader. We show that we can retain the singular values, thus decreasing the sparsity of the covariance matrix diagonal and fully utilizing the capacity of the latent dimension (we also added Sections 3.2 and 3.3 introducing the c(VAE) and PU-Net). This results in lower Gini indices and overall better model performance.
>
> **The paper cites some self-supervised learning papers, but I don't know why they relate to the target task in this paper, the models and goals are completely different.**
>
> We would like to direct the reviewers attention to the third paragraph of the Introduction. Here, we have added details about the resemblance between the goals of the PU-Net and JEMs in SSL. Furthermore, we discuss some more details about this in Section 2.2. In short, both networks attempt to learn the embedded mutual information of the two inputs that differ representationally but not semantically.
>
> **Both Figure 2 and equation 12 are not clear. For example, the paper denotes $Q$ as a posterior network, so what does $Z\sim Q$ means? Or "the sample after NF" as $\hat{Q}$ or $\hat{P}$ what does it mean?**
>
> We use ‘$\sim$’ to indicate sampling. Thus, $Z\sim Q$ implies sampling $Z$ from distribution $Q$. The "sample after the Normalizing Flow" implies a sample after being subject to the NF transformations.
>
> **Also, what is the role of normalizing flow here?**
>
> The Normalizing Flow allows us to project the probabilistic latent spaces as is done in previous work in JEMs with SSL. With an NF, we are able to weigh the individual samples when performing the Sinkhorn iterations. We have added this description at the end of of the third paragraph in the Introduction.
>
> **Figure 2 shows the distribution is based on one data, but in section 3.2.1 they define the Wasserstein distance between aggregation posterior and prior.**
>
> We learn the aleatoric uncertainty within one datapoint. In the context of segmentation, this implies the variability of segmenting. Thus, the aggregated prior and posterior are per data point. We have addressed this point in the first paragraph of Section 3.3.
>
> **The planar flow mentioned in section 3.1 is not bijective without further constrain.**
>
> We have included it in the revision of the paper.
>
> **Section 3.2 looks like a sloppy updation from other sources, it is hard to believe that section 3.2 and section 4 are written by the same person.**
>
> The Theory section is using past work to justify using the Wasserstein distance in latent space instead of the KL divergence, in case the prior is not fixed to $\mathcal{N}(0,\mathbf{I})$. It is, in our opinion, crucial to the paper. We would like the reviewer to comment in more detail what makes the section sloppy, so we can improve this.
>
> **I recommend a strong rejection of the paper. I think the author should rewrite the paper by making the following points clear to make it a valid paper before sending it to review.**
> - **Formally describe your task and give an introduction to the conditional VAE method.** The workings of a (c)VAE/PU-Net are introduced in additional sections.
> - **Describe the singular value vanishing problem in the introduced task and models.** We have added section 3.2.2 for this
> - **Give intuition about how to alleviate the problem.** To the best of our knowledge, we have introduced a problem in the PU-Net that has not been discussed before. We attempt to solve the vanishing of singular values via an adapted method that has empirically shown to work previous research in deterministic models.
> - **Use the right math notation to describe the proposed method and why it can help alleviate the problem.** We understand if some notation was confusing. We have changed and fixed some notation that will hopefully clear up confusion.

---

> > ### Comment · Reviewer_coKj · 2022-10-09
> > **Reply to the authors**
> >
> > We thank the author for the replies. However, the revised paper doesn't address my concerns.
> >
> > Most of the equations about the ELBO are wrong (e.g. 3a-3d,7), the ELBO is a lower bound of the log-likelihood, not a density value. For example, equation 3a should be $\log p(y,\phi)\geq ...$, and if you want to  integrate over $p(y)$, the LHS is $\int \log p(y,\phi)p(y)dy$.  There are also many other sloppy mistakes, for example, the LHS of equation 6 should be conditional distribution.
> > Additionally, the mathematical notations in the paper are inconsistent across different sections, which makes the paper impossible to follow.
> >
> > >We have updated our paper with an additional Section (3.4)
> >
> > I didn't find section 3.4 in the revised paper, have the authors submitted the correct version?
> >
> > I think the paper is not a qualified scientific paper yet, I strongly recommend the authors rewrite the paper before submitting it to any venues.

---

> > > ### Author Response · Authors · 2022-10-10
> > > **Reply to the coKj**
> > >
> > > Thank you for the reply. Our comment is supposed to refer to section 3.2.2 of the paper, our apologies. Also, we acknowledge our confusing notation and will carefully rewrite the manuscript.

---

### Review · Reviewer_LbnM · 2022-08-23

**Summary Of Contributions:**

In this paper, the authors conjecture that the performance of existing probabilistic U-Net approaches can be improved by mitigating the dimensional collapse of latent representations, where dimensional collapse is signaled by vanishing singular values. This paper extends existing frameworks by using the normalizing flow to represent both prior and posterior distribution over the latent variables. Additionally, the authors use ideas from optimal transport to augment the training objective in addition to already used KL and reconstruction loss.

**Broader Impact Concerns:**

.

**Requested Changes:**

Please refer to the strength and weakness section.

**Strengths And Weaknesses:**

# Strengths

This paper address an important problem of improving the methods that quantify aleatoric uncertainty in the image segmentation tasks. The authors take on an interesting direction of using normalizing flow to enhance the latent space representation. Overall, I think the idea is very natural and promising.

# Weakness

### Unclear Writing
The current manuscript needs revision. The authors sometimes go into too many details and references without getting to the point. For example, on page one, "Although this model has been..." the authors go on to explain what work has been done with NFs and PU-Net without clearly stating what the issue Is (there is only conjecture: "we have found that it does not capture ambiguity.")

Moreso, the entire introduction reads like a related work section without clearly articulating what the problem is with using normalizing flows with PU-Net and how they resolve it.

Further, a lot of math has been thrown in the theory section. However, I doubt any of this is a novel contribution. Instead, it seems to me there is an attempt for mathification of the paper.

### Naive Baselines
If I am not mistaken, one possibility is to use the KL divergence between the posterior and prior densities learned by the flows. Can the authors comment on this? It seems it can provide a baseline with the same number of parameters and modeling capacity, with the only difference originating from the objective function.

### Claims
Overall, the claim of this paper seems to be that dimensional collapse is the source of the inferior performance of PU-Net. However, I did not find any experiments demonstrating this was the case (I could not understand much from figures 3 and 4; it does not seem to me that the singular values are vanishing for any of the other methods.) Overall, it is unclear how exactly the authors mitigate the vanishing singular value problem with their formulation.


### Singular Values
The literature suggests vanishing singular values related to the intrinsic dimensionality of the problem (see discussion in [1]). However, I am unconvinced if forcing the use of all dimensions of the representational space is as crucial as the authors seem to hypothesize. Can the authors comment on this?

1. Bin Dai and David P. Wipf. Diagnosing and enhancing VAE models. ICLR, 2019.
2. Bin Dai and David P. Wipf. Diagnosing and enhancing VAE models. [Extended] https://arxiv.org/pdf/1903.05789.pdf.

---

> ### Author Response · Authors · 2022-09-26
> **Reply to reviewer LbnM 1/2**
>
> We would like to thank the reviewer for the feedback and ideas. We feel our paper was thoroughly studied. We are appreciative of the efforts and agree with the majority of points made. Regarding the weaknesses:
>
> **The authors sometimes go into too many details and references without getting to the point … Moreso, the entire introduction reads like a related work section without clearly articulating what the problem is with using normalizing flows with PU-Net and how they resolve it.**
>
> In retrospect, we indeed see this to be the case. We have revised the introduction to be more concise and truly introductory to the problem. Also, references to related work will be minimized in this section and will be referred to in its due section.
>
> **A lot of math has been thrown in the theory section. However, I doubt any of this is a novel contribution. Instead, it seems to me there is an attempt for mathification of the paper.**
>
> We find the theory Section to be essential for the paper. First, we introduce the VAE and the associated problems with it (this is after our revision). Then, we attempt to replace the KL-divergence in the latent space. To justify this, we provide a very concise mention of previous works and their theorems. It is noteworthy that these works are for traditional VAEs, in contrast to our model where the prior density is learned. This has major implications, as we have discussed in Section 3.3. In our opinion, it is crucial to include this in the main content for the clarity and completeness of the paper.
>
> **If I am not mistaken, one possibility is to use the KL divergence between the posterior and prior densities learned by the flows.**
>
> This is infeasible if one uses the objective that manifest by substitution of Eq (1). We have briefly stated this in the first paragraph of the Methods Section. In short, we know from Eq. (1) that the likelihood of a sample from the posterior equals
>
> $$
> \mathrm{log}\ q_\theta (z_q\vert x, y)=\mathrm{log}\ p_{\mathcal{N}\_q} (z_{q0})-\mathrm{log} \sum \vert \det (J_{z_{q0}\rightarrow z_q})\vert .
> $$
>
> where  $\mathcal{N}\_q$ indicates the Normal density from the posterior network. A sample $z_q$ evaluated on $p_\phi$ is its likelihood on $\mathcal{N}_p$ (from the prior network) minus the log jacobian determinant resulting from the prior flow as
>
> $$
> \mathrm{log}\ p_\phi (z_q \vert x)=\mathrm{log}\ p_{\mathcal{N}\_p} (z_{q})-\mathrm{log}\sum \vert \det (J_{z_q\rightarrow z_q'})\vert .
> $$
>
> Substituting these equation in the ELBO results in the objective
>
> $$
> \mathrm{ELBO_{NF}}=\mathbb{E}\_{q_\theta (z\vert x, y)}\mathrm{log}\ p_\psi (y\vert z_q, x) - \mathrm{KL}\left[\mathcal{N}\_q \vert\vert \mathcal{N}\_p\right] + \left(\mathrm{log}\sum \vert \det (J_{zq0\rightarrow z_q})\vert - \mathrm{log}\sum \vert \det (J_{z q \rightarrow z_q'})\vert \right)
> $$
>
> Matching likelihoods and optimizing this objective does not guarantee samples being close in latent space. Furthermore, minimizing the negative ELBO will result in the term $\mathrm{log}\sum \vert \det (J_{z_q\rightarrow z_q'})\vert$ to grow exponentially negative leading to an instability in training, as there is no mechanism countering the localization of $z_q'$.
>
> In addition, we evaluate by Hungarian-matching. This is in a sense a discrete form of the Wasserstein distance. In this manner, our objective during training matches the evaluation metric. Hence, it makes more sense to use the Wasserstein distance in latent space.
>
> **The claim of this paper seems to be that dimensional collapse is the source of the inferior performance of PU-Net. I did not find any experiments demonstrating this was the case (I could not understand much from figures 3 and 4; it does not seem to me that the singular values are vanishing for any of the other methods.**
>
> In Table 1 and Figure 3, it can be seen that the cSAE has lower Gini indices. This means that the latent space is less sparse and thus the singular values are not vanishing. Table 2 (Table 5 after revision) and Figure 4 show outlier cases for a dataset that is significantly different to that of the data in Table 1 and Figure 3. We have added Section 3.2.2 with additional details on the singular values. We have also made some additions on the Gini index and its connection to the vanishing of singular values in Section 4.2.

---

> > ### Author Response · Authors · 2022-09-26
> > **Reply to reviewer LbnM 2/2**
> >
> > **It is unclear how exactly the authors mitigate the vanishing singular value problem with their formulation.**
> >
> > Our work is inspired from JEMs with contrastive learning in SSL. There, using a projection layer on the embedding spaces empirically improved the latent representation (singular values of the latent vectors were retained). We note that the PU-Net also suffers from vanishing singular values and it serves as a significant bottleneck. We propose a solution that implements projection layers (the NFs) compatible with probabilistic latent spaces. We have made more explicit mention of this connection. Therefore, we would like to refer the reviewer to the third paragraph of the Introduction and Section 2.2 of the revised paper.
> >
> > Supporting theory on why the singular values exactly vanish are limited to a linear neural networks [[1](https://arxiv.org/abs/2110.09348)] as deep non-linear networks are infeasible to analyze. Nevertheless, it is empirically shown that the problem occurs in large non-linear networks.
> >
> > **I am unconvinced if forcing the use of all dimensions of the representational space is as crucial as the authors seem to hypothesize.**
> >
> > Before evaluation on the test set, the latent dimensionality was optimized. We can therefore assume that $r=d$, where $d$ is the latent space dimensionality and $r$ the embedded manifold of annotation variability. It has been shown [[2](https://arxiv.org/abs/1903.05789), cited by reviewer] that the posterior network collapses to low-noise variances in the traditional VAE in order to perfectly reconstruct the input. This can cause a severe mismatch between the aggregated posterior and prior $\mathcal{N}(0,\mathbf{I})$ with fixed variance. In our case, we do not want perfect reconstructions by definition of the aleatoric uncertainty. Furthermore, the prior variance is learned and some dimensions are able to collapse with the posterior. We show this happens by means of the Gini index of the singular value vector $\vec{\sigma}$. Thus, during testing, we sample from a low-variance prior distribution that can not fully represent the inter-observer variability (=aleatoric uncertainty) in the data. We then use an alternative objective and empirically show this increases the prior latent space variances, reducing sparsity of $\vec{\sigma}$. We then show that this significantly improve test evaluation.
> >
> > Regarding the cited paper provided by the reviewer, we have added Section 3.2 and 3.3 discussing its relevance to our work. In our case, the prior distribution singular values are heavily influenced by the posterior distribution. While in the traditional VAE, the prior distribution remains unchanged from initialization. This poses a different challenge than that of the conventional VAE.

---

### Review · Reviewer_oppp · 2022-09-13

**Summary Of Contributions:**

This paper proposes Conditional Sinkhorn Auto-Encoders (cSAEs), an extension to the existing Sinkhorn Auto-Encoders (SAEs) that allows to learn conditional distributions. In addition to adding conditioning to the model, the authors also use Normalizing Flows (NFs), both on the prior and approximate posterior of the model, which they argue helps avoid vanishing singular values of the learned covariance on latent space. The authors then use their model for the task of image segmentation, where the goal is to learn the conditional (on an image) distribution of pixel annotations. Overall, while I believe this paper presents a sensible though not particularly insightful methodology for conditioning in SAEs which seems to slightly outperform the considered baselines (PU-Net and its NF-augmented variant, 2p-planar), I found the empirical confirmation of the relevance of non-vanishing singular values as a key factor for performance, as well as the mathematical presentation, to both be lacking.

Before continuing the review, I will point out that while I am very familiar with generative models, including Sinkhorn Auto-Encoders and NFs, I am not as familiar with image segmentation. I do not know how standard the datasets used in the experiments are, nor what typical performance is on those datasets, nor how strong the baseline the authors compare against is considered in the literature. While I do have opinions (detailed below) about the empirical performance of cSAEs, I have a larger degree of uncertainty about these opinions than about those related to the rest of the paper.

**Broader Impact Concerns:**

I have no broader impact concerns.

**Requested Changes:**

- Please address point 3 above, both from a conceptual and an empirical standpoint [required].
- Please clean up the mathematical presentation [required].
- The phenomenon of vanishing singular values is very related to posterior collapse in VAEs. Posterior collapse has been studied before and linked to the intrinsic dimension of the observed data [B], could you please comment/discuss connections? [not required]

[B] Diagnosing and Enhancing VAE Models, Dai and Wipf, ICLR 2019

**Strengths And Weaknesses:**

# Strengths

1. The idea for conditioning SAEs is simple and sensible, and while not particularly novel (as it really is analogous to how other generative auto-encoder models enable conditioning, e.g. conditional variational auto-encoders), I see it as a positive to enable more generative models with the ability to model conditional distributions.
2. The proposed method seems to have good empirical performance over the baseline: Table 1 shows cSAEs consistently outperforming PU-Nets and 2p-planar across different metrics and datasets, and albeit the obtained improvement is not outside error bounds for most if not all the metric/dataset pairs, the fact that cSAEs outperform the baselines across all such pairs does suggest they work better.

# Weaknesses

3. One of the main points of the paper is that as compared to competing baselines, cSAEs have fewer vanishing singular values in the covariance matrix on latent space, and that as a result of this, their performance is improved. I do not think the paper provides enough support for this claim, neither at a conceptual level, nor empirically. At a conceptual level, it is easy to see that the considered singular values are not invariant to reparameterizations. To see this, consider a Gaussian prior with diagonal covariance obeying that no singular value is vanishing (i.e. very close to 0), which is then fed into the NF component. Changing this prior by scaling each element of the Gaussian appropriately, one can clearly make any of the singular values vanish, yet the scaling can be undone by the NF. This change would result in a different prior, which now has vanishing singular values, yet would produce the same generative model overall, highlighting that non-vanishing singular values are not necessarily behind good empirical performance. On the empirical side, I also have some issues. First, the use of the Gini index as a measure of sparsity of the singular value vector is somewhat surprising to me, as I tend to think of the Gini index more as a diversity measure. While the authors do cite a paper to motivate this use of the Gini index (Hurley and Rickard, 2009), it would be good to provide some additional intuition as to why this is indeed a good metric. For example, the 'effective rank' (i.e. number of singular values above a pre-specified threshold) seems at first glance as a more natural way of measuring vanishing singular values. Second, part of the argument presented by the authors is that the addition of NFs to the model is part of what enables fewer vanishing singular values, which in turn increases performance. From Table 2, we can see that 2p-planar has fewer vanishing singular values than its non-NF counterpart, PU-Net (as measured by the Gini index), yet the performance improvement from PU-Net to 2p-planar is either extremely marginal or nonexistent, suggesting that the non-vanishing singular values are not necessarily the reason why cSAEs perform well. Third, in the same vein as the second point, there is no empirical comparison against the cSAE version that forgoes NFs. The lack of this ablation makes it difficult, once again, to be convinced by the author's claim that non-vanishing singular values are indeed behind the observed performance gains.

4. I found the mathematical presentation to lack clarity, and I believe I was able to follow only because I am very familiar with Wasserstein Auto-Encoders and SAEs, but I strongly suspect any reader only vaguely familiar with these methods would find the presentation confusing and miss at least some details (maybe I did so inadvertently). While there is no single huge issue in the presentation, I find that all the small issues do add up in a relevant manner. Below I list all these issues in order of appearance:
- In section 3.1, $\mathbf{x}$ is defined as living in $\mathbb{R}^{D \times D}$, which is inconsistent with equation 2, which suggests that $\mathbf{x} \in \mathbb{R}^D$. Also, $f_i$ should map from $\mathbb{R}^D$ to $\mathbb{R}^D$ according to equation 2, not from $\mathbb{R}$ to $\mathbb{R}$ as defined before equation 1.
- Also in Section 3.1, $\mathbf{x}$ is defined as Gaussian, yet if it is obtained by transforming a Gaussian $\mathbf{z}_0$ through the NFs, it need not be Gaussian.
- In section 3.2.1, $(X,Y)$ is said to be in $\mathbb{R}^{D \times D} \times \mathbb{R}^{D \times D}$, which implies that $Y \in \mathbb{R}^{D \times D}$. I believe the considered algorithm does not require $Y$ to take scalar values.
- In section 3.2.1, $\mathcal{P}$ is not defined either, and the phrasing 'we define $P_Z$ on $\mathcal{P}$' is very poor, as it implies that $\mathcal{P}$ is the sample space, rather than a set of distributions containing $P_Z$, as becomes clear later.
- In equation 4, $\mathcal{X}$ (which has not been defined) and $\hat{\mathcal{Y}}$ are used, but presumably these are both just $\mathbb{R}^{D \times D}$ according to the first line of section 3.2.1.
- In equation 4, the density $p_G(\hat{y}|z,x)$ is written down, with the text 'assuming all involved densities are properly defined'. The authors then turn their attention to the case where $G$ is deterministic, which implies that 'density' is a point mass and thus not well defined.
- In the definition of $W_c$ in equation 5, the set $\mathcal{P}_{Y, \hat{Y}}$ is used without ever being defined.
- In the discussion following equation 5, $P_G$ is mentioned without being defined, and $\mathcal{Q}$ is also not defined.
- When defining the encoding distributions $Q_Z$ before theorem 1, it is actually the marginals that are being defined. This is quite confusing, as the encoding distributions, as defined, are not conditional on neither $X$ nor $Y$. I think it is important to emphasize that the object being optimized over are the conditional distributions $Q(Z|X,Y)$, with a constraint on their implied marginal $Q_Z$. This is repeated throughout the entire mathematical presentation: it is often ambiguous what different distributions are conditioned on (e.g. in equation 7 there is also a lot of ambiguity).
- The notation $W_p$ in equation 8 is poor, as it appears to be $W_c$ with $c=p$, rather than a different quantity altogether.
- In equation 10, I believe there is a '$d$' missing before entropy term, yet the Radon-Nikodym derivative $d \gamma / d\mu d \nu$ need not be well defined in general (e.g. if $\mu$ is a standard 1-dimensional Gaussian, $\nu=\mu$, and $\gamma$ is obtained by sampling $Y$ from $\mu$, and then setting $\hat{Y}=Y$). In the same vein, the authors mention the 'coupling matrix' right before equation 10, which assumes the problem is discrete, which need not happen at the level of generality being presented in the paper (as defined in section 3.2, $\mathcal{Y}$ and $\hat{\mathcal{Y}}$ are taken as arbitrary separable metric spaces). I understand that this ends up being used to compare empirical measures which are discrete, but again, the mathematical exposition is not precise.
- In section 4, the use of $\mathbf{c}$ for the amortized parameters of the NF are again a poor notational choice, as $c$ is already used as the cost function used to define the Wasserstein distance. Similarly, equation 12 uses $S_{c, \epsilon}$, when only $S_{\epsilon}$ has been defined in equation 11.
- In equation 13, the definition of $\mathcal{Y}$ and $\hat{\mathcal{Y}}$ has been changed from metric spaces to observed samples. Similarly, couplings between sets of samples $\Gamma(\mathcal{Y}, \hat{\mathcal{Y}})$ are not defined.
- The letter $\mathbf{c}$ is further overloaded in equation 16 onwards.

Some additional minor points/typos:
- 'enabls learning of the aleatoric uncertainty' -> 'enables learning the aleatoric uncertainty'
- Figure 1 is never referenced in the text.
- 'Rezende Rezende'
- I think [A] should be cited in section 2.3.
- 'the intricacies of the Wasserstein distance complicates its calculation' -> 'the intricacies of the Wasserstein distance complicate its calculation'
- The paper formatting is rather unsightly, pages 9 and 10 have a lot of blank space.

[A] Wasserstein Auto-Encoders, Tolstikhin et al., ICLR 2018

---

> ### Author Response · Authors · 2022-09-26
> **Reply to reviewer oppp 1/2**
>
> We want to thank the reviewer for the extensive comments and feedback. The reviewer's expertise is appreciated and we are convinced the constructive criticism will significantly improve our work. Regarding the baseline and datasets, we can claim with certainty that the PU-Net is the most popular choice for probabilistic image segmentation with follow-up research being published every year since it has initially been proposed (see second paragraph of the introduction). Although other methods for aleatoric uncertainty quantification in segmentation exist, few have been practically adapted due to the various drawbacks as stated in the original PU-Net paper. Regarding the dataset, we can say with confidence that the LIDC-IRDI dataset has become the standardized task for aleatoric uncertainty quantification. The other datasets we have used are not as popular, likely because of their limited size.
>
> We will answer the posed questions and concerns per point.
>
> **At a conceptual level, ... highlighting that non-vanishing singular values are not necessarily behind good empirical performance.**
>
> Indeed, a vanishing singular value is not invariant to reparameterization by the NF during training. However, during testing, we remove the NF and sample from the axis-aligned Normal densities. If, hypothetically speaking, a singular value of the prior distribution vanishes but is compensated by the NF, this will hurt evaluation during testing because we are sampling from the collapsed Normal densities. We have added this clarification in the first paragraph of the Methods Section. Note, that this practice is identical to the non-probabilistic projection MLP layers in Joint Embedding Methods (JEMs) in self-supervised learning, and results similarly in retained singular values on the embeddings *before* the projection MLP, which is consequently utilized for the downstream task.
>
> **On the empirical side, ... it would be good to provide some additional intuition as to why this is indeed a good metric.**
>
> We have implemented the effective rank in our revised paper as this indeed seems to be the more intuitive metric. From our understanding, the effective rank is entropy based. According to (Hurley and Rickard, 2009), entropy based metrics are sub-optimal measures for sparsity (although the effective rank itself is not experimented with). Therefore, we had chosen the Gini index, since it adhered to all six criteria for measuring sparsity.
>
> **Second, part of the argument presented ... suggesting that the non-vanishing singular values are not necessarily the reason why cSAEs perform well.**
>
> During revision of the paper, we have found an systematic error in our evaluation scripts. This was related to how we handled correct empty predictions (which are returned as NaNs by the IoU and Dice). We have fixed the issue but want point the reviewer to the fact that the ranges of the numbers are now slightly different.
>
> Except for the augmented NFs, all experiments and model architectures were identical during training. The only difference observed is the significant difference in the singular values of the latent densities. As an effect, we see that this improves test evaluations most of the times. Very occasionally, the difference is indeed marginal to non-existent. To us, this does not negate the positive effect of the retained singular values. Rather, this indicates that for that particular dataset, the relatively smaller singular values were not the bottleneck during testing performance. After thorough investigation and given previous work on the VAE, the latent space sparsity seems the most likely explanation for improved segmentation performance.
>
> **Third, in the same vein as the second point ... makes it difficult, once again, to be convinced by the author's claim that non-vanishing singular values are indeed behind the observed performance gains.**
>
> We have not been able to successfully generalize a model that was trained on purely the Sinkhorn iterations. We have found the KL-divergence to be an essential component. For future work we intend to build a model that can solely converge with the Sinkhorn iterations (or with MMD, as this has been used in other works) and see how this affects the latent space.

---

> > ### Author Response · Authors · 2022-09-26
> > **Reply to reviewer oppp 2/2**
> >
> > **While there is no single huge issue in the presentation, I find that all the small issues do add up in a relevant manner. Below I list all these issues in order of appearance.**
> >
> > We want to thank once again for the thorough review. The detailed feedback has enabled us to improve the presentation and hopefully remove any confusion in our revised paper. We are eager to know if we have misunderstood any feedback provided by the reviewer.
> >
> > **The phenomenon of vanishing singular values is very related to posterior collapse in VAEs. Posterior collapse has been studied before and linked to the intrinsic dimension of the observed data [B], could you please comment/discuss connections?**
> >
> > We have added discussions relating to [B] in Sections 3.2 and 3.3.

---

> > > ### Comment · Reviewer_oppp · 2022-10-06
> > > **Discussion**
> > >
> > > I thank the authors for their reply. I have re-read the updated manuscript, and a lot of my original concerns remain:
> > >
> > > 1. `Indeed, a vanishing singular value is not invariant to reparameterization by the NF during training. However, during testing, we remove the NF and sample from the axis-aligned Normal densities. If, hypothetically speaking, a singular value of the prior distribution vanishes but is compensated by the NF, this will hurt evaluation during testing because we are sampling from the collapsed Normal densities. We have added this clarification in the first paragraph of the Methods Section. Note, that this practice is identical to the non-probabilistic projection MLP layers in Joint Embedding Methods (JEMs) in self-supervised learning, and results similarly in retained singular values on the embeddings before the projection MLP, which is consequently utilized for the downstream task`
> > >
> > > I still do not think removing the NFs shows that the vanishing singular values were indeed causing performance issues. Rather, it shows that one can get a well-performing model that has no vanishing singular values, yet if one re-added an NF, one could obtain an equally well-performing model that does not have this property. In other words, I do not think the relationship between non-vanishing singular values and the model working is causal as suggested in the manuscript.
> > >
> > > 2. As for the presentation, I still find that while it has been slightly improved, the paper remains mostly hard to follow, and I believe a more thorough re-write of the paper is in order. I will also point out that, while [B] is now mentioned, it is not discussed in the context I was expecting (and that I suspect reviewer LbnM was also expecting): I don't think mentioning the paper in the context of background on VAEs is particularly insightful, rather I think it would be interesting to link the results from [B] about posterior collapse with the claims being made about vanishing singular values of latent representations.

---

> > > > ### Author Response · Authors · 2022-10-10
> > > > **Reply to reviewer oppp**
> > > >
> > > > Thank you for your reply. We mention in the Methods section: “Furthermore, this design allows for the NFs to eventually converge to identical densities. In other words, initially, during training the projected space is far from the base Normal densities. As training continues, the projected latent spaces coincide with the base Normal densities”. Thus, keeping or discarding the NFs makes close to no difference with regards to the singular values of the latent samples. Perhaps more attention should be given to this statement. In retrospect I am also convinced this could have been stated better.
> > > >
> > > > As for the second point, we have discussed the results of [B] in context of the PU-Net. From [B] it was clear that the aggregated posterior is ill-matched with the prior Eq. (4), because the collapsed posterior. We then state in Section 3.2.1 that the aggregated prior and posterior can be well-matched, because the prior is not a fixed, but rather a learned density. However, this introduces the issue where the prior can collapse together with the posterior and have a limited latent representation (i.e. low singular values).
> > > >
> > > > Nevertheless, we acknowledge the presence of confusing notation and lack of conceptual reasoning. We will rewrite our manuscript and pay special attention to both. Again, special thanks to the reviewer for being thorough and detailed in his response.

---

### Comment · Action_Editors · 2022-11-27
**Reject decision**

Dear Authors
We thank you for your rebuttal and discussion with reviewers. They all unanimously lean towards rejection. The paper is still not prime for a publication, and I encourage the authors to take into account comments to continue polishing further their draft. Additionally, please pay close attention to the various remarks on the content itself of the paper, which, in its current scope, may not be sufficient to grant publication @ TMLR.
AE

---

### Decision · Action_Editors · 2022-11-27

**Recommendation:** Reject

**Comment:**

All reviewers agree on the fact that this submission does not meet the bar for TMLR. Writing needs several more rounds of polishing, and ideas are tentative. Experimental evidence is preliminary. This is a paper that was clearly submitted too early.

**Audience:**

At this moment, the paper is not well written enough, even to touch specialized audiences. A future revision of this paper might.

**Claims And Evidence:**

The paper has received a fairly negative round of reviews, All 3 reviewers recruited for this task have expressed frustration at the lack of clarity of the paper itself (in its form), but also on the hypothesis formulated by the authors and its lack of convincing empirical demonstration.